# Experimental Investigation: Vibration Measurement of a Rotating Blade with Digital Image Correlation and Blade Tip-Timing

**DOI:** 10.3390/mi13122156

**Published:** 2022-12-06

**Authors:** Zhonghan Liang, Yuxiang Zhang, Lin Yue

**Affiliations:** College of Mechanical and Electrical Engineering, Nanjing University of Aeronautics and Astronautics, Nanjing 210016, China

**Keywords:** digital image correlation, blade tip-timing, dynamic strain, blade vibration

## Abstract

High cycle fatigue has been known as an important form of aeroengine blade failure. This study aims to achieve a method of investigation for a rotating blade vibration measurement, combining the two non-contact optical techniques of digital image correlation (DIC) and blade tip-timing (BTT). Dynamic parameters of a thin-blade were obtained on a stationary vibration platform with stereo-DIC system. Meanwhile, the finite element analysis (FEA) of this thin-blade was performed within different rotating speeds. Then, the set of thin-blades was mounted in a simulated compressor test rig equipped with BTT and a wireless strain gauge (SG) system. A rotor speed sweep experiment was carried out and the blade synchronous resonance parameters were extracted. Results show that the displacement mode shapes match well between DIC and FEA, and that MAC values of the first six order modes are over than 0.88. The predicting strain from the FE model and SG agreed to within 32.41% in the worst case, and the predicting strain from the DIC model corresponds to 28.53% in the worst case. This is an effective non-contact, high-precision full-field deformation measurement method that is worth exploring for structural design and dynamic strain assessment of vibrating components.

## 1. Introduction

High cycle fatigue has been known as an important form of aeroengine blade failure. Vibration in turbomachinery could reduce the blades’ fatigue life by increasing the risk of crack formation. The blade health monitoring represents an important challenge in order to prevent unexpected blade failures, and the two tasks of health monitoring and life estimation are undertaken. The rotor blade vibration is detected using the strain gauges (SG) with a slip-ring, which still represents the most reliable measurement system nowadays. However, the engine in service cannot be equipped with it due to the measurement principle of contact pattern [1,2].

For this reason, a well-known non-contact measurement technique, blade tip-timing (BTT), is currently employed for the identification of the dynamic behaviors of rotating blades and structure health monitoring (SHM). In the most applications, the BTT sensors are mounted on the casing and oriented towards the blade tips [3]. The latest generation industry-standard BTT systems process the arrival times by using indirect [4,5] or direct [6,7] identification methods to determine the modal parameters such as modal frequencies and amplitudes. A common method for predicting dynamic strain is to construct a finite element (FE) model of the blade and solve the dynamic modal parameters. Although this method is powerful and very useful, modeling errors (geometry, boundary conditions, damping, etc.) may result in inaccurate strain prediction.

Some proposed research has demonstrated a good correlation between the BTT and strain gauge measurements for both real and controlled operation conditions [8,9]. The blade root and trailing edge are often the regions where conventional dynamic strain prediction uses strain-gauges to gather a few identified strain responses, since they have larger strain amplitudes in relation to one another. However, the position of the few strain-gauges can only roughly cover the desired location due to the complex loads on the thin-wall parts. Therefore, a full-field displacement and strain measurement is urgently needed. Engine development heavily relies on rig tests, which need the creation of improved measuring techniques. Existing predictive methods for either synchronous or asynchronous vibration have not attained the degree of precision needed for design calculations.

Digital image correlation (DIC) is another non-contact optical measurement technique, which takes advantage of full-field deformation tracking with high accuracy up to 0.01~0.02 pixels [10,11,12,13]. Recently, a dozen published papers [14,15,16] were focused on the displacement mode shapes of non-rotating blades or thin-plate models using the high-speed DIC (HS-DIC), with a few endeavors toward the strain mode shapes measurement [17,18,19]. In addition, a few experimental studies on rotating blades have been conducted [18], but they are only capable of measuring displacement modes under the conditions of low rotational speed and large-radius with low linear velocities and low natural frequencies. For instance, a 2 m diameter helicopter rotor blade was subjected to the operational modal analysis (OMA) using an HS-DIC system by S. Rizo-Patron et al., and the first three flap bending nature frequencies and mode shapes were measured and extracted within 900 rpm [20]. D. Uehara et al. also described the 2 m diameter coaxial counter-rotating (CCR) rotor blade three-dimensional deformations with HS-DIC method [21]. Actually, the larger radius of the rotating blade, the higher linear velocity of the blade tip in conditions of the same synchronous vibration rotational speed. Helicopter rotor blades are much larger than compressor blades, and that is a challenge to the hardware performance of high-speed cameras in DIC measurement for the same algorithm capability. Generally, the displacement in pixel units during exposure is one of the most critical HS-DIC system parameters for dynamic measurement. From the displacement mode shapes to the strain mode shapes, and from the non-rotating dynamic model to the rotating dynamic model, these are the two critical challenges for the current application of aeroengine blade health monitoring. Essentially, it is also limited by current HS-camera performance in spatial resolution and temporal resolution. There have been attempts in the literature to examine displacement mode shapes with rotating blade and strain mode shapes with non-rotating blade [17,19,20]. Most of these attempts were studied on the metal or composite material thin-plate, with emphasis on the DIC measurement quality. This is rarely trialed for strain mode shape applications, especially in a rotational rig. Additionally, laser Doppler vibrometer (LDV) is another kind of non-contact optical vibration measurement technique and some work combining a laser Doppler vibrometer (LDV) and BTT for dynamic strain reconstruction was also implemented recently [22,23]. Lots of earlier work has been carried out to compare DIC and LDV measurement results and they each have their own strengths [24,25,26,27,28].

This piece of work tries to combine the BTT and DIC techniques to achieve a non-contact rotating blade vibration measurement. The accuracy of the FE model is critical to dynamic strain prediction and high cycle fatigue issues. The HS-DIC could achieve full-field dynamic displacement and strain measurement which could establish a more accurate FE model with full-field experimental data. In current work, ratio *k* of the blade strain to the blade tip displacement was defined for dynamic strain prediction model and the experiment verified the effectiveness of this method.

## 2. Methodology

As an optical measurement technology, DIC uses digital cameras to obtain the grayscale image which contains the information of the displacement field on the surface of the structure concerned. Generally, monochrome cameras are adopted to obtain the high-quality grayscale images, rather than complex interpolated color cameras. Black and white speckles are sprayed or transferred, printing on the structural surface to enhance a high contrast and random speckle pattern. The optimized and recommended speckle size is 3~5 pixels [29]. Then, the region of interest (ROI) in the reference image is divided into small subsets for enough gray-scale information for a correlation algorithm. As shown in Figure 1, there is a squared subset in the reference image corresponding to a deformed subset in the target image. (u,v) represents the displacement of the subset center and a series of nonlinear iterative algorithms [30,31,32] aimed at solving it. Cross-correlation and the sum of squared differences are the most-used definitions of the correlation criteria. Commonly, the zero-normalized cross-correlation (ZNCC) and the zero-normalized sum of squared differences (ZNSSD) are shown as Equations (1) and (2), respectively. The equation (2 M + 1) represents the subset size in pixel units. In Equations (3) and (4), $f_{m}$ and Δf represent the mean value and mean square deviation of subset pixel gray in the reference image, $g_{m}$ and Δg corresponding to the deformed image. ZNCC and ZNSSD were proved to be equivalent, and ZNSSD was selected in this work. This is a brief description of local DIC, and for more details please refer to some early literature [11,13]. The sub-pixel level displacement tracking is the core process of DIC algorithm, and it has been developed for more than two decades. The main methods include, but are not limited to, the Gray–Gradient method [33], the Newton–Raphson method [34], and the IC–GN method [35]. For a more detailed description, please refer to Section 3 of the DIC algorithm in [36], and the Appendix D of the classic book [10] in DIC research. Our in-house DIC code was based on the Newton–Raphson method mentioned earlier, and the current popular method is the IC–GN method. Essentially, this is a multi-dimensional nonlinear locally optimal optimization problem. It involves shape functions, gray-scale interpolation, and non-linear optimization methods.

As described, the subset-based DIC method applies the correlation interpolant to a set of independent smaller subsets of the images and the resulting displacements are only defined over a finite set of interrogation points, which are point-wise and iterative calculated; the so-called local-DIC method. In contrast, a global-DIC method performs on an a priori chosen interpolant based on a set of shape functions to represent and solve the displacement field over the entire ROI of images. Local- and stereo-DIC methods were adopted in this work. For details of stereo-DIC, please refer to [10,13], which involves stereo-calibration and three-dimension reconstruction omitted here.
(1)$$C_{ZNCC}=\sum_{i=-M}^{M}\sum_{j=-M}^{M}\left\{\frac{\left[f(x_{i},y_{j})-f_{m}]\times [g({x^{\prime }}_{i},{y^{\prime }}_{j})-g_{m}\right]}{\Delta f\Delta g}\right\}$$
(2)$$C_{ZNSSD}={\sum_{i=-M}^{M}\sum_{j=-M}^{M}\left[\frac{f(x_{i},y_{j})-f_{m}}{\Delta f}-\frac{g({x^{\prime }}_{i},{y^{\prime }}_{j})-g_{m}}{\Delta g}\right]}^{2}$$
(3)$$\begin{array}{l} f_{m}=\frac{1}{{\left(2M+1\right)}^{2}}\sum_{i=-M}^{M}\sum_{j=-M}^{M}f(x_{i},y_{j}),\\ g_{m}=\frac{1}{{\left(2M+1\right)}^{2}}\sum_{i=-M}^{M}\sum_{j=-M}^{M}g({x^{\prime }}_{i},{y^{\prime }}_{j}). \end{array}$$
(4)$$\begin{array}{l} \Delta f=\sqrt{\sum_{i=-M}^{M}\sum_{j=-M}^{M}{\left[f(x_{i},y_{j})-f_{m}\right]}^{2}},\\ \Delta g=\sqrt{\sum_{i=-M}^{M}\sum_{j=-M}^{M}{\left[g({x^{\prime }}_{i},{y^{\prime }}_{j})-g_{m}\right]}^{2}}. \end{array}$$

The schematic diagram of a typical BTT system and pulse signal sequence are shown in Figure 2 and Figure 3, respectively. Tip-timing is the measurement of blade tip vibration using non-contact optical or capacitance probes located around the casing. The BTT sampling data are inherently under-sampled and contaminated with several measurement uncertainties [37]. The rotating blade’s time of arrival (TOA), containing both blade vibration and the assembly’s rotational speed, is measured at each of the blade passing probes. An additional once per revolution (OPR) probe is used to record the rotational speed. The blade tip deflection in rotational direction is estimated from the TOA and rotational speed data. As is shown in Figure 3, $T_{n}$ represents the period of *nth* cycle between two adjacent OPR pulses. $t_{nij}$ and $t_{nij0}$ represent the TOA with vibration and TOA without vibration based OPR pulse, respectively. It is defined that the blade following close after the OPR pulse is the first blade. The subscripts *i* and *j* represent the blade number and the probe number. For this diagram, the blade number *i* is 9. Then, in Equation (5), the blade vibration displacement $y_{nij}$ is derived from the difference between $t_{nij}$ and $t_{nij0}$, transient rotational period $T_{n}$, and blade tip rotational radius *r*.
(5)$$y_{nij}=2\pi r\cdot \left(t_{nij}-t_{nij0}\right)/T_{n}$$

According to these, a series of vibration displacement response $y_{nij}$, the blade’s dynamic characteristics can be determined through further data post-processing algorithms [1,3,38,39,40,41]. For instance, a fitting method named circumferential Fourier fit (CFF) [33] was adopted in this work, which requires four or more probes. That is the value range of a positive integer j no less than four, assuming each blade is vibrating according to a sinusoidal wave with a certain response engine order (EO) [3,7]. The blade motion function is shown in Equation (6), where $\theta_{j}$ is the angular position of *jth* probe relative to the OPR marker on shaft and its fixed parameters for a certain BTT system. Then, for a certain EO, the amplitude $A\left(\omega \right)$, phase $\Phi \left(\omega \right)$, and zero drift DC are calculated through a least square method with the several probes’ response data.
(6)$$\begin{array}{c} y_{j}\left(\omega \right)=A\left(\omega \right)\cdot \sin\left(\mathrm{EO}\cdot 2\pi ft+\Phi \left(\omega \right)\right)+\mathrm{DC}\\ =A\left(\omega \right)\cdot \sin\left(\mathrm{EO}\cdot \theta_{j}+\Phi \left(\omega \right)\right)+\mathrm{DC} \end{array}$$

Additionally, a dynamic strain system is used to valid the results from the BTT system and a set of wireless strain gauge systems was adopted in this experimental investigation as shown in Figure 2. In Equation (7), *k* is defined to describe the ratio of blade root strain *ε* to the blade tip displacement *u*. Correspondingly, $k_{\mathrm{mod}}$, $u_{\mathrm{mod}}$, and $\varepsilon_{\mathrm{mod}}$ are the parameters in modal space. In following comparative analysis, *k* defaults to the ratio of modal parameters, where $k_{\mathrm{FE}}$ and $k_{\mathrm{DIC}}$ represent the results from FEA and DIC, respectively. The predicting dynamic strain ${\bar{\varepsilon }}_{\mathrm{DIC}}$ and ${\bar{\varepsilon }}_{\mathrm{FE}}$ are defined in Equation (8), and are validated through the wireless strain-gauge results of $\varepsilon_{\mathrm{SG}}$. As is shown in Figure 4, a DIC experimental model was integrated in the original method and used for full-field dynamic strain prediction. With the help of abundant full-field experimental data, an HS-DIC method is expected to model validation, full-field response expansion and other applications in future.
(7)$$k_{\mathrm{mod}}=\varepsilon_{\mathrm{mod}}/u_{\mathrm{mod}}$$
(8)$${\bar{\varepsilon }}_{\mathrm{DIC}}=u_{\mathrm{BTT}}\cdot k_{\mathrm{DIC}}\;{\bar{\varepsilon }}_{\mathrm{FE}}=u_{\mathrm{BTT}}\cdot k_{\mathrm{FE}}$$

## 3. Experimental Set-Ups

### 3.1. Stereo-DIC Experimental Setups

This is a standard high-speed stereo-DIC experiment in line with the guidebook from iDICs. As shown in Figure 5a, two high-speed cameras were fixed on a tripod, Phantom V1212. The widescreen CMOS monochrome sensor can acquire and save up to 12 Gigapixels/s of data, at full megapixel resolution of 1280 × 800 @12,000 fps. Two Nikon AF-S NIKKOR 85 mm f/1.8 G lenses were matched. For illumination, a strong light LED with maximum power of 500 watts was adopted, equipped with 25 ultra-bright 5 mm beads and adjustable light intensity. Short exposure and small aperture setting can be ensured by such sufficient light intensity. The laser displacement sensor was used to feedback vibration amplitude information in a single frequency excitation test, to adjust a series of accurate resonance frequencies from the output of the signal generator to the vibration actuator.

As is shown in Figure 5b, the thin-blade was clamped onto the vibration platform and a layer of water transform speckle was spread on the upper surface. Total dimensions of the rectangular metal thin-blade were 138 × 80 × 0.5 mm^3^ and the depth of the clamping part was 18 mm as a fixed support boundary. Thus, the net size of the cantilever was 120 × 80 × 0.5 mm^3^. Speckle patterns were made by a Speckle Generator, including these three main parameters of diameter, density, and variation. In this work, the diameter was set as 0.8 mm and the density and variation were set at the recommended value of 75%. A pair of original reference images and their gray-scale histograms of ROI are shown in Figure 6. In Figure 6b, the locally enlarged speckle patterns show that the diameter of speckle dot is around five pixels. Meanwhile, the gray-scale histograms of ROI display the gray value distribution, mainly ranging from 15 to nearly 230, which widely indicates satisfied speckle quality and appropriate setting combinations of aperture, shutter, object distance, light intensity, etc.

Considering the natural frequencies of this thin-blade range from tens to hundreds of Hertz, the camera frame rate was set as 2000 fps and the shutter time was 30 μs for enough depth of field (DOF). Acquisition time lasted about 5 s, and nearly 10,000 images were taken from each camera for the subsequent DIC calculation in which the grid spacing was set as 5 pixels and the square subset size was set as 31 pixels. First order shape function was adopted due to there being no large deformation in this case. The correlation matching process is based on the Ncorr, which is an open source 2D DIC tool [36]. The stereo-vision calibration process was carried out through the MATLAB tools in this work. Then, we developed an in-house code with MATLAB to achieve 3D reconstruction and DIC follow-up sessions.

### 3.2. BTT Experimental Setups

As is shown in Figure 2 and Figure 7, this BTT rig contains 12 thin-blades and a set of wireless strain gauges attached on the number 4 and number 5 blades. In Figure 8a, two strain gauges were attached on each blade at the blade root and middle. Thus, there were four strain channels connected to the wireless transmitter in total. The sampling rate of strain data acquisition was set as 1000 Hz. Meanwhile, four fiber laser probes were embedded in the rotor-case and the optimized circumferential angle distribution was 30°, 70°, 148°, and 222°. An NI PXI-6612/6602 timer/counter was used as data acquisition hardware, with the internal clock frequency up to 80 MHz.

Another critical structural parameter is the blade tip rotational radius *r* of 300 mm, which includes the blade net height of 120 mm. In Figure 8b, considering the blade stagger angle *θ* ≈ 10.8°, the vibration displacement measured from BTT system is not equal to the real blade vibration displacement. Equation (9) illustrates the relationship between them, where *u*_vib_ represents the vibration displacement in a normal direction and *u*_measured_ represents the vibration displacement measured in a circumferential direction. It is worth noting that the variable *u*_BTT_ in Figure 4 refers to *u*_vib_ here. When the blade arrives at the probes, the distance is around 17 mm to the leading edge of the blade. Then, a group of speed up and down tests were conduct and the rotating speed reached 800 rpm. Four EO and three EO blade synchronous resonances were excited and recognized.
(9)$$u_{vib}=u_{measured}\times \cos\theta$$

## 4. Results and Analysis

With the help of *Workbench* in Ansys, modal analysis of this thin-blade was also simulated in this work. It is necessary to demonstrate some material parameters, firstly, such as the density, $\rho =7.93\times 10^{3}$ kg/m^3^; Young’s modulus, E=200 GPa; and Poisson’s ratio ν=0.3. The solid shell element dimension was 1 mm. A total of 9801 nodes on the thin-blade were extracted in the modal analysis, as is shown in Figure 9. The boundary condition was unilateral fixation, such as a cantilever plate, and the dimensional parameters are shown in Figure 5b.

### 4.1. DIC Results and Validation with FE Model

Previous works have used the same thin blade to demonstrate the displacement precision of HS-DIC for a number of single points in the time domain and frequency domain, respectively [42,43]. The results of DIC data processing in modal space after full-field displacement are as follows.

In this work, the displacement modes of DIC results were from a hammer excited experiment. Then, the poly-reference least-squares complex frequency-domain method (p-LSCF) [44] was employed for modal parameters identification. The stable chart is shown as in Figure 10 and the poles of the first six order of displacement modes were selected. Correspondingly, the first six order displacement mode shapes in the out-of-plane direction of the FEM and DIC results were presented together.

Due to the thin-blade being only 0.5 mm thick and the influence of internal stress, there are still some differences in the frequencies within a reasonable range. For instance, the elevated ellipse in the fifth order of the displacement mode shape has a different long axis orientation in the DIC and FEM results. Despite all this, the MAC values of these first six order displacement modes are more than 0.88. This shows that the traditional way, which depends on the FE model directly, is risky due to modeling errors, especially for the thin-wall part.

The strain modes of the DIC results were from the single frequency excited experiment through the shaker with a set of control systems, because this is a small strain challenge task of ten microstrain, and the signal noise ratio (SNR) was raised. Due to the rotational speed limit, only the first order mode of this thin-blade was excited and recognized in the BTT experiment. Thus, the first bending mode frequency at 28.47 Hz was excited through the shaker. Correspondingly, the stable chart and strain mode shapes are shown in Figure 11, in which the strain tensor was written as $E_{xx},E_{yy},E_{xy}$. As expected, full-field mode strain of $E_{xx}$ and $E_{xy}$ are close to zero. The main strain components are concentrated in $E_{yy}$ which is in the same direction with strain gauges, especially in the blade root. Meanwhile, the spectral lines with several multiple frequencies were unexpected and considered as from the shaker system.

Moreover, the Campbell diagram was also obtained with FEM in Figure 12. The nature frequencies change slightly when the rotating speed is less than 800 rpm. The effect of rotational stiffness hardening and softening is not readily apparent due to the low speed. The two dotted lines are three EO and four EO, which are close to the first order dynamic frequency curve. The speed of blade synchronous resonance at crossover points and the ratio *k* of blade tip displacement to blade strain are shown in Table 1, and the nodes are extracted at the same positions as the measurement points shown in Figure 8a. *k*_root_ and *k*_middle_ represent the ratio *k* of two different strain gauge positions.

### 4.2. Comparisons of the Dynamic Strain at the First Order of Mode Frequency

The ratio *k* from FEM is displayed in Table 1, and then the findings from the DIC and BTT experiment are explored. The DIC results were from a single frequency excitation at the first order mode, which is the first bending mode, and the interested positions of displacement and strain in frequency domain are shown in Figure 13, where the positions are consistent with the BTT measurement as is shown in Figure 8a. Further, these response amplitudes were gathered in Table 2, and the ratio $k_{\mathrm{DIC}}$ was calculated according to the Equation (7). Comparing with Table 1 and Table 2, the ratio *k* is clearly consistent and the DIC results are a little small; that is, considering that they are caused by strain measurement noise, such as the influence of strain window filters at the blade root with large strain gradient. On the other hands, $k_{\mathrm{FE}}$ also contains lots of uncertainties, such as the inconsistency of local boundary conditions and geometric shape errors.

In the BTT measurement, the blade synchronous vibrations of four EO and three EO were identified. The tip displacement amplitudes $A\left(\omega \right)$ were identified through the CFF method with Equation (6) and represented by $u_{\mathrm{BTT}}$ to keep consistent with the DIC results. The strain amplitudes were extracted through the short-time Fourier transform (STFT) method with Hanning windows and the frequency resolution was set as 1 Hz. Due to the blade synchronous vibration time being less than 1 s, the strain amplitudes errors were brought from the STFT process. It was necessary to select an appropriate frequency resolution to keep the balance of enough frequency accuracy and unaveraged vibration amplitudes when the rig passed through rotational speed of the blade synchronous vibration. In Table 3, firstly, the displacements $u_{\mathrm{BTT}}$ and strains $\varepsilon_{\mathrm{SG}}$ of the three EO blade synchronous vibration were gathered. Then, the predicting strains ${\bar{\varepsilon }}_{\mathrm{FE}}$ and ${\bar{\varepsilon }}_{\mathrm{DIC}}$ were derived through Equation (8) and the $k_{\mathrm{DIC}}$ and $k_{\mathrm{FE}}$ were selected from Table 1 and Table 2. The relative errors followed closely. All these strains from Table 1, Table 2, Table 3 and Table 4 were gathered in Figure 14, which makes the comparison clearer. The predicting strain from the FE model and SG agreed to within 32.41% in the worst case, and the predicting strain from the DIC model corresponds to 28.53% in the worst case. Moreover, ratios of *k* were also gathered in Figure 15, which indicated the difference between different models.

## 5. Conclusions

This is an effective non-contact, high-precision full-field deformation measurement method that also is a way that is worth exploring for the structural design and dynamic strain assessment of vibrating micromachine components. The test scale depends on the field of view (FOV) of the imaging system. Displacement mode shapes are matching well between DIC and FEA. The dynamic strain of the three EO and four EO blade synchronous vibration are validated and basically consistent. Meanwhile, this is a potential route for blade detection, strains assessment, and health monitoring in turbomachinery. The DIC capability of full-field displacement and strain measurement could be used for FE model correction and validation, which can be a benefit for the BTT system in blade health monitoring. Especially for the thin-wall part, for example of the fan blade and rotor case, it is very challenging to establish accurate FE model directly. This work is the first attempt to combine BTT and DIC methods aiming at the rotational blade dynamic response measurement and prediction. A DIC experiment was carried out on platform in this work. Essentially, the influence of the rotational speed on dynamic stress and strain cannot be ignored after the speed rises for most rotors. On the one hand, the speed influence is added to the static corrected FE model with DIC experiment; on the other hand, it is worth trying to measure the dynamic response of the rotating blade with stereo-DIC system and validating to the dynamic FE model directly. This is a challenge to the DIC hardware testing capability, in temporal and spatial resolution, and it is likely to be accompanied by a small strain challenge of ten microstrain as in this work. With the help of abundant full-field experimental data, the HS-DIC method is expected to model validation, full-field response expansion and other applications in microscale in future.

## Figures and Tables

**Figure 1 micromachines-13-02156-f001:**
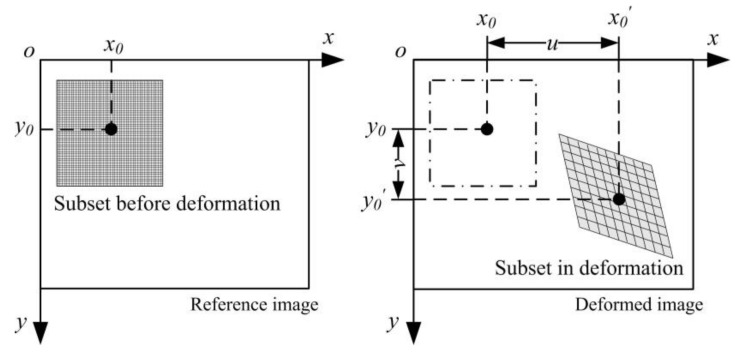
Schematic diagram of the subset in local DIC method.

**Figure 2 micromachines-13-02156-f002:**
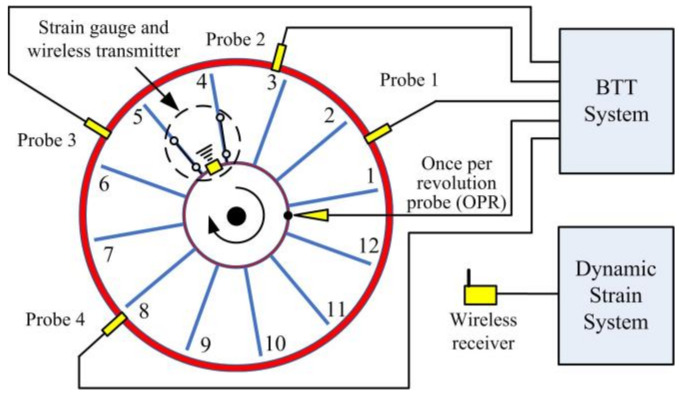
BTT and dynamic strain system diagram.

**Figure 3 micromachines-13-02156-f003:**
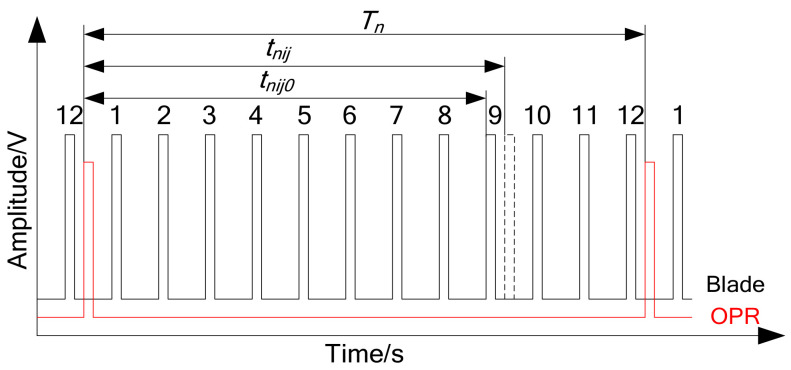
Pulse signal sequence for one blade passing probe and once per revolution probe.

**Figure 4 micromachines-13-02156-f004:**
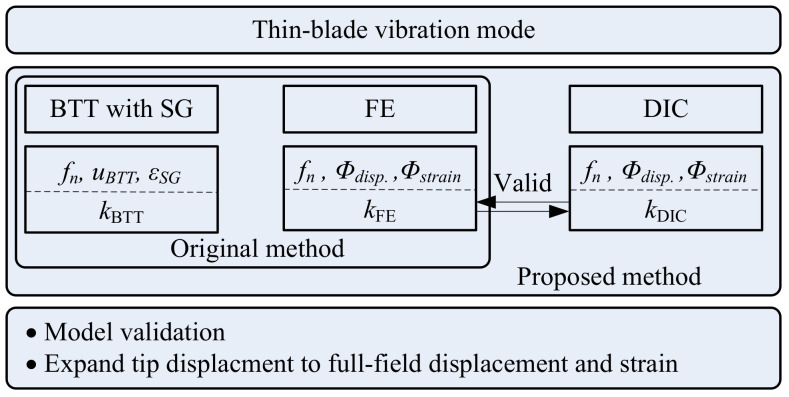
Methodology diagram.

**Figure 5 micromachines-13-02156-f005:**
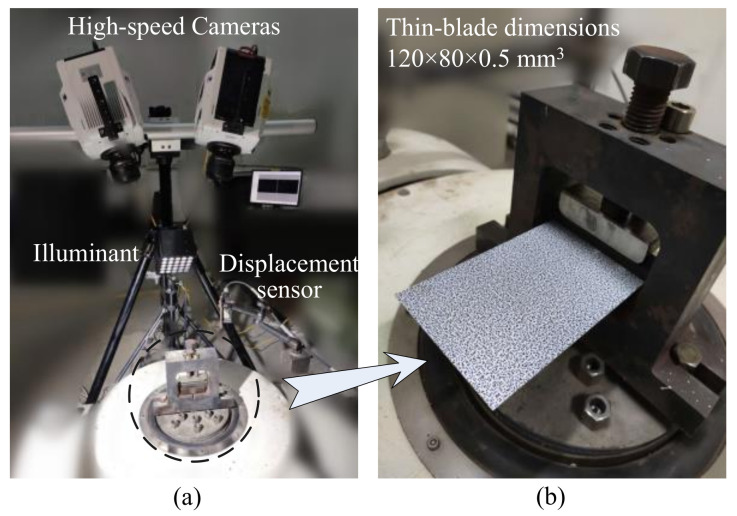
Stereo-DIC set-ups: (**a**) High-speed DIC system; (**b**) The thin-blade covered with speckle patterns.

**Figure 6 micromachines-13-02156-f006:**
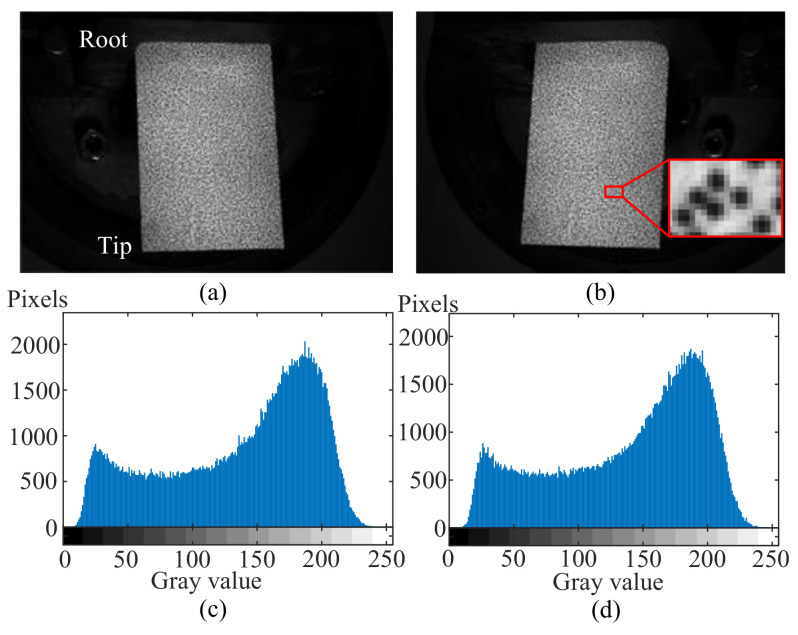
A pair of original reference images from high-speed cameras and the gray-scale histogram of speckle patterns: (**a**) Left image; (**b**) Right image; (**c**) Gray-scale histogram of the ROI in left reference image; (**d**) Gray-scale histogram of the ROI in right reference image.

**Figure 7 micromachines-13-02156-f007:**
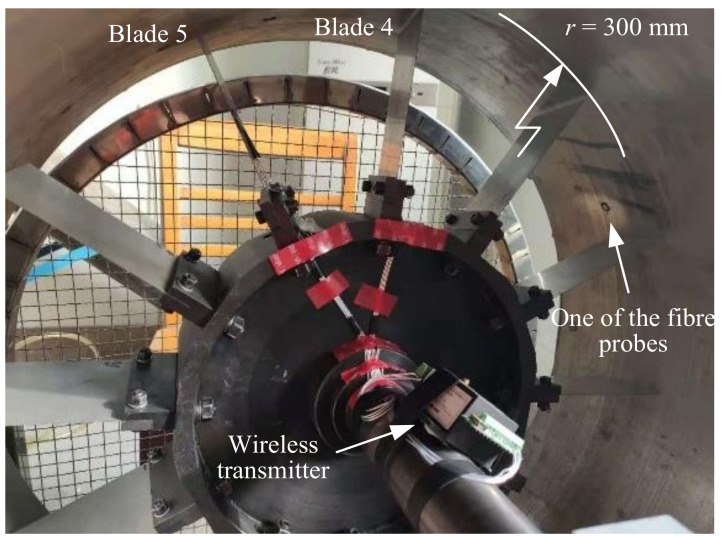
BTT test rig with wireless strain gauges.

**Figure 8 micromachines-13-02156-f008:**
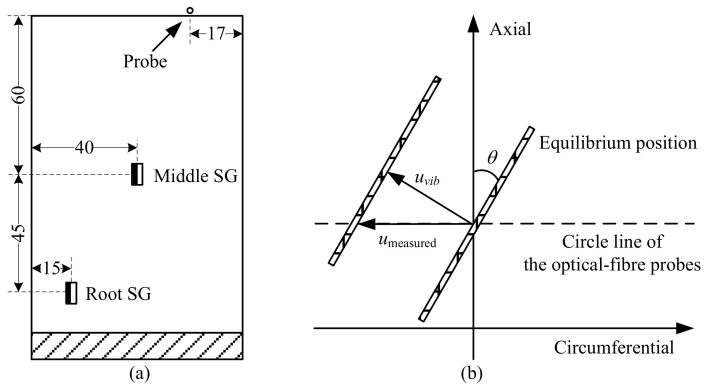
Schematic of strain gauge positions and blade set-ups: (**a**) Positions of middle SG and root SG; (**b**) Blade stagger angle *θ*.

**Figure 9 micromachines-13-02156-f009:**
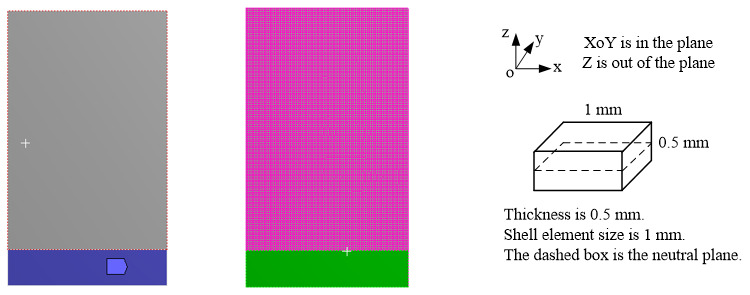
FE model of this thin blade and the schematic of one shell element.

**Figure 10 micromachines-13-02156-f010:**
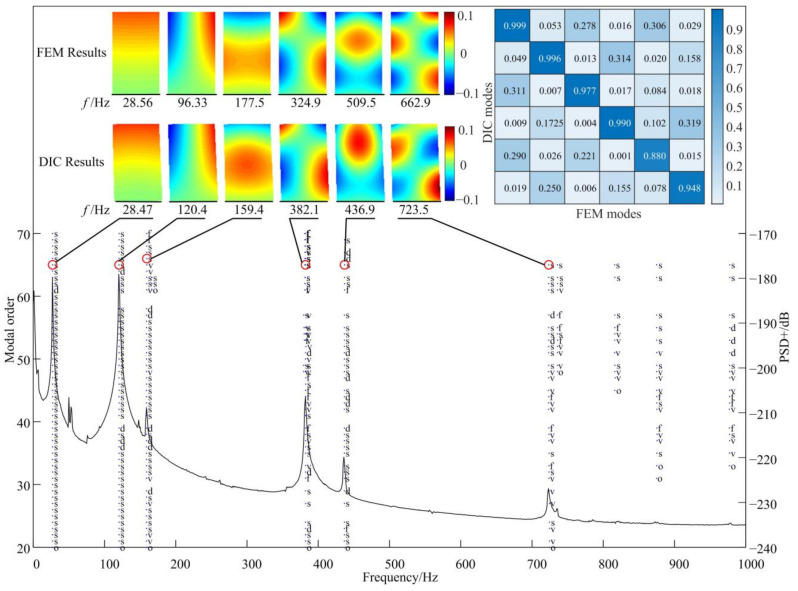
Stable chart and displacement mode shapes in out-of-plane direction.

**Figure 11 micromachines-13-02156-f011:**
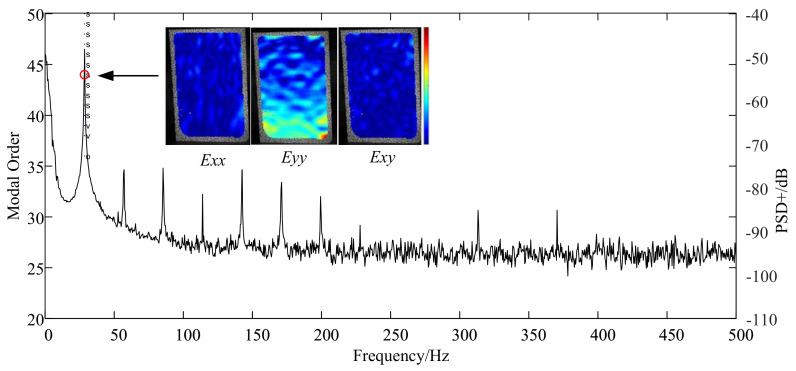
Stable chart and the first order of strain mode shapes.

**Figure 12 micromachines-13-02156-f012:**
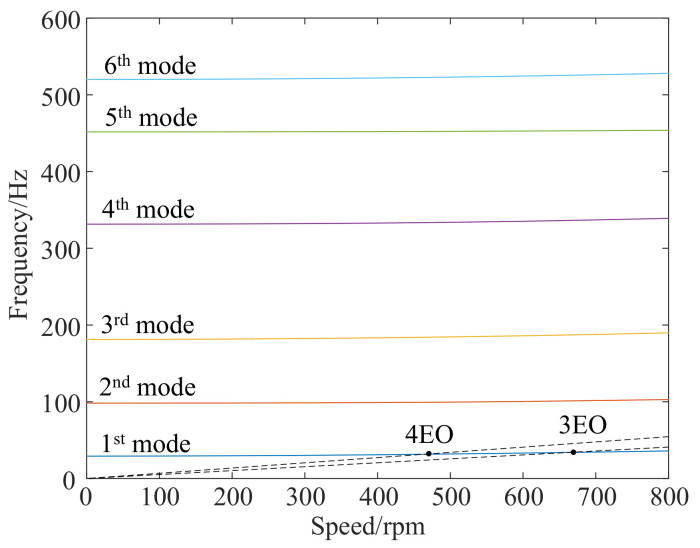
Campbell diagram of FEM.

**Figure 13 micromachines-13-02156-f013:**
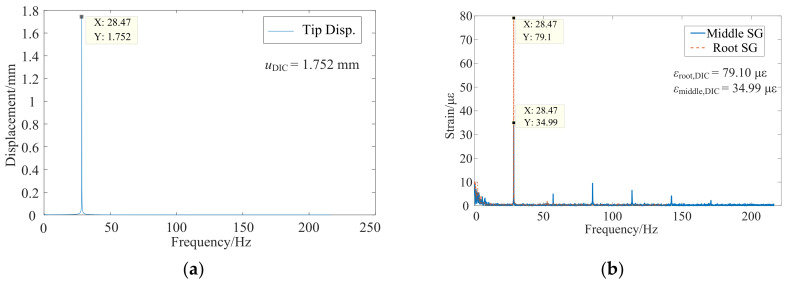
DIC results of interested displacement and strain in frequency domain under a single frequency excitation: (**a**) Blade tip displacement in frequency domain; (**b**) Blade middle and root strain in frequency domain.

**Figure 14 micromachines-13-02156-f014:**
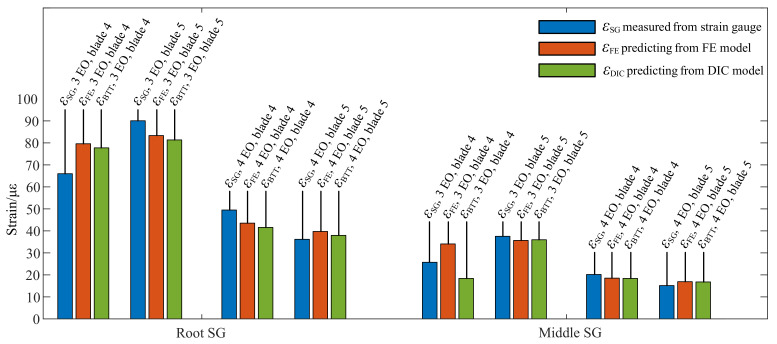
Comparative of measured strains and predicting strains the first order of mode frequency.

**Figure 15 micromachines-13-02156-f015:**
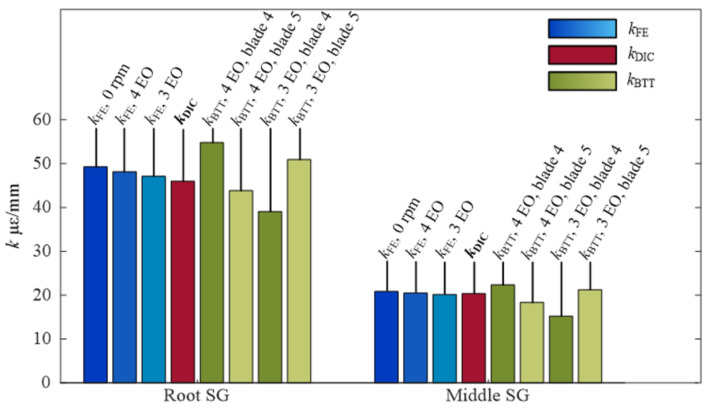
Ratio *k* of blade strain to tip displacement at the first order of mode frequency.

**Table 1 micromachines-13-02156-t001:** Ratio *k* in different blade synchronous vibration crossover points of FEM.

$k_{\mathrm{FE}}\;\mu \varepsilon /\mathrm{mm}$	No Speed	4 EO @ 479.7 rpm	3 EO @ 682.8 rpm
$k_{\mathrm{FE},\mathrm{root}}$	49.27	48.17	47.13
$k_{\mathrm{FE},\mathrm{middle}}$	20.85	20.50	20.17

**Table 2 micromachines-13-02156-t002:** Ratio *k* from the DIC experiment at the first order of mode frequency.

$u_{\mathrm{DIC}}\mathrm{mm}$		$\varepsilon_{\mathrm{DIC}}\mu \varepsilon$		$k_{\mathrm{DIC}}\;\mu \varepsilon /\mathrm{mm}$		$\mathrm{Relative}\;\mathrm{Errors}\;\mathrm{to}\;k_{\mathrm{FE}}$
$u_{\mathrm{tip}}$	1.752	$\varepsilon_{\mathrm{DIC},\mathrm{root}}$	79.10	$k_{\mathrm{DIC},\mathrm{root}}$	46.02	6.59%
		$\varepsilon_{\mathrm{DIC},\mathrm{middle}}$	34.99	$k_{\mathrm{DIC},\mathrm{middle}}$	20.35	1.12%

**Table 3 micromachines-13-02156-t003:** Dynamic strains from SG and predicting model in blade synchronous vibration of 3 EO.

$u_{\mathrm{BTT}}\;\mathrm{mm}$		$\varepsilon_{\mathrm{SG}}\;\mu \varepsilon$		${\bar{\varepsilon }}_{\mathrm{FE}}\;\mu \varepsilon$		Relative Errors	${\bar{\varepsilon }}_{\mathrm{DIC}}\;\mu \varepsilon$		Relative Errors
Blade 4	1.6892	$\varepsilon_{\mathrm{SG},\mathrm{root}}$	65.99	${\bar{\varepsilon }}_{\mathrm{FE},\mathrm{root}}$	79.61	20.64%	${\bar{\varepsilon }}_{\mathrm{DIC},\mathrm{root}}$	77.74	17.81%
		$\varepsilon_{\mathrm{SG},\mathrm{middle}}$	25.73	${\bar{\varepsilon }}_{\mathrm{FE},\mathrm{middle}}$	34.07	32.41%	${\bar{\varepsilon }}_{\mathrm{DIC},\mathrm{middle}}$	18.39	28.53%
Blade 5	1.7678	$\varepsilon_{\mathrm{SG},\mathrm{root}}$	90.05	${\bar{\varepsilon }}_{\mathrm{FE},\mathrm{root}}$	83.32	7.47%	${\bar{\varepsilon }}_{\mathrm{DIC},\mathrm{root}}$	81.35	9.66%
		$\varepsilon_{\mathrm{SG},\mathrm{middle}}$	37.54	${\bar{\varepsilon }}_{\mathrm{FE},\mathrm{middle}}$	35.66	5.01%	${\bar{\varepsilon }}_{\mathrm{DIC},\mathrm{middle}}$	35.97	4.18%

**Table 4 micromachines-13-02156-t004:** Dynamic strains from SG and predicting model in blade synchronous vibration of 4 EO.

$u_{\mathrm{BTT}}\;\mathrm{mm}$		$\varepsilon_{\mathrm{SG}}\;\mu \varepsilon$		${\bar{\varepsilon }}_{\mathrm{FE}}\;\mu \varepsilon$		Relative Errors	${\bar{\varepsilon }}_{\mathrm{DIC}}\;\mu \varepsilon$		Relative Errors
Blade 4	0.9035	$\varepsilon_{\mathrm{SG},\mathrm{root}}$	49.49	${\bar{\varepsilon }}_{\mathrm{FE},\mathrm{root}}$	43.52	12.06%	${\bar{\varepsilon }}_{\mathrm{DIC},\mathrm{root}}$	41.58	15.98%
		$\varepsilon_{\mathrm{SG},\mathrm{middle}}$	20.18	${\bar{\varepsilon }}_{\mathrm{FE},\mathrm{middle}}$	18.52	8.23%	${\bar{\varepsilon }}_{\mathrm{DIC},\mathrm{middle}}$	18.39	8.87%
Blade 5	0.8250	$\varepsilon_{\mathrm{SG},\mathrm{root}}$	36.18	${\bar{\varepsilon }}_{\mathrm{FE},\mathrm{root}}$	39.74	9.84%	${\bar{\varepsilon }}_{\mathrm{DIC},\mathrm{root}}$	37.97	4.95%
		$\varepsilon_{\mathrm{SG},\mathrm{middle}}$	15.14	${\bar{\varepsilon }}_{\mathrm{FE},\mathrm{middle}}$	16.91	11.69%	${\bar{\varepsilon }}_{\mathrm{DIC},\mathrm{middle}}$	16.79	10.90%

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
