# Peer review of "Experimental Investigation: Vibration Measurement of a Rotating Blade with Digital Image Correlation and Blade Tip-Timing"

_micromachines, 2022, doi:10.3390/mi13122156_

Round 1

Reviewer 1 Report

[1] Writing is poor which makes it difficult to follow. This is true throughout the article. Serious editing and proofreading are required. 

[2] The authors talk about MEMS systems but performed analysis on larger size components, i.e. thin plate of 120mm in length. Why is that? If possible demonstrate the methodology on MEMS-scale object. Otherwise, modify the introduction section mentioning all kinds of similar work carried out on different kinds of blades including turbine and helicopter blades. 

[3] Did the authors develop an in-house DIC code or use open-source/commercial DIC software? Please indicate and explain accordingly. 

[4] What is the basis for the statement "The optimized and recommended speckle size is 3~5 pixels" (Line 116-117)? Explain with a proper reference. 

[5] The authors mentioned two correlation methods, ZNCC and ZNSSD. How were they used?  Did you employ both or one of them? 

[6] Please describe the correlation process in more detail. Also, explain how the sub-pixel level accuracy was achieved. 

[7] The method of printing speckle patterns is interesting. Please show the corresponding experimental setup. 

[8] Which part of the ANSYS tool was used for FE analysis? Show the computational FEA model of the plate using an image having the necessary details. 

Reviewer 2 Report

In this paper, a method investigation of a rotating blade vibration measurement is proposed, which combines the two non-contact optical techniques of digital image correlation (DIC) and blade tip-timing (BTT) at the macroscale. A set of thin-blades are mounted in a simulated compressor test rig which equipped with BTT and wireless strain gauges (SG) system. Rotor speed sweep experiment is carried out and the blade synchronous resonance parameters were extracted. Finally, some conclusions are made by the authors. However, the reviewer thinks that there are still some questions needing to be answered before it can be published.

1.In Section 4: which element is selected for blade finite element model? Is the thickness 1mm (line 245) or 0.5mm(line 196)?

2.In 3.2, line 221: How are the strain measuring points of blade selected? Please describe of the BTT system including hardware and software, sampling frequency.

3.In 4.1, fig 9: In the test results of DIC, the relative error value of the second order natural frequency and finite element value is more than 20%. It is quite large. Please explain it.

4.In 4.2, line 323: The relative error of 3EO is generally larger than that of 4EO. Please explain this phenomenon.

5.In 4.2, table 2&table3,4: In Table 2, when comparing the relative error between displacement-strain transfer ratio k calculated by FEM and DIC method, the ratio k with speed is used, however, when comparing the predicted strain calculated by the two methods in Table 3 and Table 4, the dynamic ratio k is used (considering the rotation effect). The unfixed comparison standard may not make the analysis process clear, which may lead that the analysis results are not convincing.

6.In 4.2, table3&4: Using DIC instead of FEM to calculate the displacement-strain transfer ratio is carried out on the vibration table, without considering the rotation effect in the process of blade rotation, so whether this will lead to errors between the strain calculation results and the actual strain measurement results of rotating blades.

7.In introduction, line 62:For the strain prediction based on BTT measurements, analytical relationship between the blade dynamic strain and tip displacement are modeled, for example, “Dynamic strain reconstruction of rotating blades based on tip timing and response transmissibility” and “Non-contact full-field dynamic strain reconstruction of rotating blades under multi-mode vibration”. The reviewer suggests authors review on these methods in the introduction section to give readers alternatives to perform dynamic strain prediction based on BTT measurements. Additionally, some work on LVD for measuring blade vibration can also be added.

8.Generally, the accuracy of measuring blade displacement using DIC is not enough high, when the small blade vibrates under low level. As the review of the introduction, DIC was used for large structure which has large displacement. Please discuss the application promising of DIC for real aero-engine.

Reviewer 3 Report

This paper attempts to combine blade tip timing (BTT) and digital image correlation (DIC) measurements to monitor the vibration response of a blade, which is validated by strain gauges and finite element analysis techniques. Experiments were performed to validate the reliability of the proposed methods.

REMARKS:

1. The logic of the full text is unclear which is difficult to understand for readers who do not have enough experience in blade vibration-related research. The author's innovation points are not expressed clearly.

2. The concept “energy harvest” are introduced in both the abstract section and the beginning of the Introduction section. However, the relationship between this concept and the theme of this paper is not obvious.

3. The description of the two measurement methods, DIC and BTT, are not clearly introduced in the Methodology section, and it is recommended to refer to "Full-field strain prediction using mode shapes measured with digital image correlation" and "Sparse Representation Based Frequency Detection and Uncertainty Reduction in Blade Tip Timing Measurement for Multi-Mode Blade Vibration Monitoring" for a more concise and explicit introduction to the principles of the two measurement methods.

4.  The Experimental set-up and Result sections only mention a set of displacement measurement results in BTT measurement. As an important measurement method mentioned in the title of this article, the results of BTT measurement should be clarified concretely from multiple perspectives, such as displacement solution method, working condition setting, measurement results and data processing. Then the accuracy of the results should be compared with other measurement results such as FEA and DIC.

6. The paper is generally well written, but should be carefully checked for typos, e.g. line 100 "attamp"; line 252 "ara".

Round 2

Reviewer 1 Report

Thanks for providing satisfactory responses to my comments. Please also incorporate them in the revised version of the manuscript, at least mention crucial details in the manuscript, explanation in the responses alone is not enough. Highlights the corresponding changes in the manuscript and mention Page No and Line No in the responses. 

Reviewer 2 Report

All the comments are well answered. It can be accepted in the current form. 

Author Response

As suggestion with Editor, we gathered all the Changes and Modifications with a short cover letter and we hope to get your approval. Please see the attachment. Thanks for your kindly help.

Reviewer 3 Report

The authors have modified the paper in detail. It looks fine and I recommend its publication in Micromachines.

Author Response

(The authors gave the same response as above.)
